# Toll-Like Receptor 7 Is Required for Lacrimal Gland Autoimmunity and Type 1 Diabetes Development in Male Nonobese Diabetic Mice

**DOI:** 10.3390/ijms21249478

**Published:** 2020-12-13

**Authors:** Ivy L. Debreceni, Michael S. Chimenti, David V. Serreze, Aron M. Geurts, Yi-Guang Chen, Scott M. Lieberman

**Affiliations:** 1Stead Family Department of Pediatrics, Carver College of Medicine, University of Iowa, Iowa City, IA 52242, USA; ivy-gandy@uiowa.edu; 2Immunology Graduate Program, University of Iowa, Iowa City, IA 52242, USA; 3Iowa Institute of Human Genetics, Carver College of Medicine, University of Iowa, Iowa City, IA 52242, USA; michael-chimenti@uiowa.edu; 4The Jackson Laboratory, Bar Harbor, ME 04609, USA; dave.serreze@jax.org; 5Department of Physiology and Cardiovascular Center, Medical College of Wisconsin, Milwaukee, WI 53226, USA; ageurts@mcw.edu; 6Department of Pediatrics, Department of Microbiology and Immunology, Medical College of Wisconsin, Milwaukee, WI 53226, USA; yichen@mcw.edu; 7Max McGee National Research Center for Juvenile Diabetes, Medical College of Wisconsin, Milwaukee, WI 53226, USA

**Keywords:** Sjögren syndrome, type 1 diabetes, Toll-like receptor 7, lacrimal gland, salivary gland

## Abstract

Sjögren syndrome (SS) is an immunologically complex, chronic autoimmune disease targeting lacrimal and salivary glands. Nonobese diabetic (NOD) mice spontaneously develop inflammation of lacrimal and salivary glands with histopathological features similar to SS in humans including focal lymphocytic infiltrates in the affected glands. The innate immune signals driving lymphocytic infiltration of these glands are not well-defined. Here we evaluate the role of Toll-like receptor (TLR) 7 in the development of SS-like manifestations in NOD mice. We created a *Tlr7* knockout NOD mouse strain and performed histological and gene expression studies to characterize the effects of TLR7 on autoimmunity development. TLR7 was required for male-specific lacrimal gland inflammation but not for female-specific salivary gland inflammation. Moreover, TLR7 was required for type 1 diabetes development in male but not female NOD mice. RNA sequencing demonstrated that TLR7 was associated with a type I interferon (IFN) response and a type I IFN-independent B cell response in the lacrimal glands. Together these studies identify a previously unappreciated pathogenic role for TLR7 in lacrimal gland autoimmunity and T1D development in male NOD mice adding to the growing body of evidence supporting sex differences in mechanisms of autoimmune disease in NOD mice.

## 1. Introduction

Sjögren syndrome (SS) is a chronic autoimmune disease that primarily targets lacrimal and salivary glands leading to progressive exocrine gland dysfunction and debilitating ocular and oral dryness [1]. The ensuing poor ocular and oral health may lead to defects in vision and difficulty with normal oral function such as talking and swallowing. Beyond these glandular manifestations, many individuals with SS develop autoimmunity affecting other organs, chronic musculoskeletal pain, and profound fatigue. The chronic immune stimulation is believed to contribute to the markedly increased risk of developing lymphoma. Current treatments largely aim to replace or augment tears and saliva but fail to adequately halt or reverse the chronic inflammation to prevent damage or progressive exocrine gland dysfunction. The lack of well-established therapeutics to modulate the immune response may relate to the gap in understanding of the early immunopathogenic mechanisms driving the exocrine gland inflammation.

Toll-like receptors (TLRs) are pattern recognition receptors that recognize a wide range of pathogen and damage associated molecules [2]. TLR signals in innate and adaptive immune cells play early roles in immune responses and have been linked to infections, cancer, and autoimmunity [3,4]. Studies of human salivary gland tissue, peripheral blood mononuclear cells (PBMCs), and mouse models of SS have identified pathogenic roles for several cell surface and intracellular TLRs in salivary gland autoimmunity [5]. TLRs are expressed at the ocular surface [6], but evidence for the role of TLRs in lacrimal gland autoimmunity is limited. Administration of a TLR3 agonist resulted in lacrimal gland inflammation in a non-autoimmune-prone mouse strain (C57BL/6) [7]. In an autoimmune-prone strain of mouse that spontaneously develops SS-like manifestations (NOD. B10Sn-*H2^b^*/J) lacrimal gland inflammation decreased when *Myd88*, which encodes a key adaptor in the signaling pathways of multiple TLRs, was disrupted [8] suggesting a pathogenic role for MyD88-dependent TLRs. TLR7 is MyD88-dependent, and a role for TLR7 was suggested by development of spontaneous lacrimal gland inflammation in autoimmune-prone mice (BXSB/MpJ-*Yaa*) in which males have increased expression of TLR7 [9].

TLR7, an endosomal TLR, has been implicated in multiple autoimmune diseases including SS and type 1 diabetes (T1D) [10,11,12,13]. TLR7 recognizes uridine-rich ssRNA and guanosine [14] and is expressed in monocytes, macrophages, dendritic cells, and B cells. Signaling through TLR7 is MyD88-dependent and uses NF-κB and interferon (IFN) regulatory factor pathways to drive expression of type I IFN and other inflammatory cytokines. In SS, *TLR7* expression was increased in PBMCs, and this increased expression positively correlated with IFN signatures in SS patients [15,16]. TLR7 has been detected in labial minor salivary glands of SS patients with expression noted in ductal cells and infiltrating immune cells [16,17]. A role for TLR7 has not been evaluated in nonobese diabetic (NOD) mice, which spontaneously develop SS-like manifestations and are a well-established model for the study of SS-like lacrimal and salivary gland autoimmunity [18].

The purpose of this study was to define the role of TLR7 in the spontaneous development of SS-like autoimmunity in NOD mice. We developed gene-edited NOD mice lacking *Tlr7* expression and found that male mice lacking TLR7 were protected from the spontaneous development of lacrimal gland inflammation, whereas females were not protected from spontaneous salivary gland inflammation. NOD mice are also a model of T1D given the spontaneous development of T1D in both sexes. In the absence of TLR7, male NOD mice were protected from T1D development, but females were not protected. Through RNA sequencing studies, we identified genes and pathways up-regulated in lacrimal glands of wild-type NOD mice compared to TLR7-deficient NOD mice and compared this gene set to a set of genes up-regulated in lacrimal gland disease in a type I IFN-dependent manner. Together, our findings suggest TLR7 plays a key pathogenic role in the development of lacrimal gland autoimmunity associated with a type I IFN response and with B cell responses in situ in a type I IFN-independent manner.

## 2. Results

### 2.1. Development of Tlr7 Knockout NOD Mice

To evaluate the role of TLR7 in the development of autoimmunity in NOD mice, we developed *Tlr7* knockout (KO) NOD mice through CRISPR/Cas9-mediated gene editing directly in NOD mouse embryos resulting in a 2 base-pair deletion in *Tlr7* (Figure 1A). *Tlr7* KO genotype was confirmed by gene sequencing, and lack of TLR7 protein in both male and female splenocytes was confirmed by intracellular flow cytometry (Figure 1B–D).

### 2.2. TLR7-Deficient NOD Mice Are Protected from Autoimmunity in a Sex-Specific Manner

NOD mice spontaneously develop autoimmunity targeting lacrimal and salivary glands (SS-like manifestations) and insulin-producing β cells in the endocrine pancreas (T1D manifestations). The SS-like manifestations occur in a sex-specific manner with males spontaneously developing lacrimal gland inflammation and females spontaneously developing salivary gland inflammation, while T1D manifestations occur in both sexes [19,20,21,22]. To determine the role of TLR7 in the SS-like autoimmune manifestations, we quantitated inflammation in lacrimal and salivary glands of 10-week-old WT and *Tlr7* KO NOD mice by standard focus-scoring. Lacrimal gland inflammation was markedly diminished with little to no inflammation detected in the glands from male *Tlr7* KO mice compared to WT male mice (Figure 2A,B). Neither WT nor *Tlr7* KO female mice developed lacrimal gland inflammation (Figure 2C,D) in accordance with the known male-specific occurrence of spontaneous lacrimal gland autoimmunity in NOD mice [19,23]. In contrast, females developed focal sialadenitis to a similar degree regardless of the presence or absence of TLR7 (Figure 2E,F). In accordance with the female-specific occurrence of spontaneous salivary gland autoimmunity in NOD mice ([20]), WT male NOD mice did not develop salivary gland inflammation (Figure 2G,H). Surprisingly, *Tlr7* KO male NOD mice developed some salivary gland inflammation (Figure 2G,H). T1D developed in most of both female and male WT NOD mice, but only male *Tlr7* KO mice were protected from T1D with 16 of 17 *Tlr7* KO male NOD mice diabetes free at 30 weeks (Figure 2I,J). While *Tlr7* is an X-linked gene that may escape X-linked inactivation [24], this does not explain the lack of protection from autoimmunity in female *Tlr7* KO mice given that the *Tlr7* KO NOD mice expressed no detectable TLR7 protein regardless of sex (Figure 1B–D). These data demonstrate a requirement for TLR7 in the development of lacrimal gland inflammation and T1D in male NOD mice but not for salivary gland inflammation or T1D in female NOD mice. Yet, the male-specific protection from autoimmunity was not global as the lack of TLR7 resulted in an increase in salivary gland inflammation in male mice. Thus, TLR7 may play more complex disease-promoting and disease-protecting functions depending on the target organ in male NOD mice, but TLR7 deficiency did not alter autoimmunity in either SS or T1D manifestations in female NOD mice.

### 2.3. RNA Sequencing of Whole Lacrimal Gland Tissue Implicates Key Immune Genes and Pathways

Given the broad possible TLR7-associated downstream signals among different immune and non-immune cells that may participate in lacrimal gland inflammation, we evaluated whole tissue gene expression patterns to identify genes and pathways that were up-regulated in lacrimal glands of WT compared to *Tlr7* KO NOD mice. We performed RNA sequencing on whole tissue RNA samples from lacrimal glands of WT and *Tlr7* KO NOD mice at ~20 weeks of age, a time when lacrimal gland inflammation was clear in WT but only minimally detected in KO NOD mice (Figure 3A). We identified 3936 significantly differentially expressed (DE) genes including 1398 genes with at least 2-fold-change between WT and KO. Of these, 1205 genes were up-regulated in WT and 193 were up-regulated in KO lacrimal glands (Figure 3B, Appendix A). Among the genes with highest differential expression up-regulated in WT were immunoglobulin genes, complement C4a, cytokines, chemokines, and other innate immunity genes (Table 1, Appendix A). To identify relevant TLR7-dependent pathways associated with lacrimal gland inflammation, we performed pathway analyses using the iPathwayGuide platform, which includes over-representation analysis of DE genes and, where applicable, incorporates the perturbation through impact analysis for pathways that have clear gene-gene interactions and directionality [25] (Table 2, Table 3 and Table 4, Appendix A). We validated the differential gene expression for several genes from those mostly highly DE and from the cytokine-cytokine receptor pathway (Figure 4).

### 2.4. TLR7 Drives Up-regulation of Type I IFN-Dependent and -Independent Genes in Lacrimal Glands

We have previously demonstrated that type I IFN signaling is required for lacrimal gland inflammation in NOD mice [23]. Given that TLR7 is a known driver of type I IFN, we wondered to what extent the gene expression profiles above reflected the lack of a type I IFN response in *Tlr7* KO mice. To address this, we compared the set of genes up-regulated in lacrimal glands of WT compared to *Tlr7* KO NOD mice (Figure 3B) with a similar set of genes up-regulated in lacrimal glands of WT compared to type I IFN signaling-deficient (*Ifnar1* KO) NOD mice [26]. Of note, these data were generated at the same time and the WT reference group is the same for both comparisons. We limited our analysis to those genes significantly DE with at least 2-fold increased expression in WT compared to the relative KO lacrimal glands. Of the 1205 TLR7-dependent genes up-regulated in lacrimal glands of WT NOD mice, 997 (83%) were also up-regulated in a type I IFN-dependent manner (Figure 5A, Appendix A). These genes were enriched for clusters of pathways and processes involved in T cell and adaptive immune responses (Figure 5B, middle). A smaller set of genes (208) was uniquely up-regulated in WT lacrimal glands in a TLR7-dependent but IFNAR1-independent manner (Figure 5A, Appendix A), and these genes were enriched for B cell responses (Figure 5B, top). Genes up-regulated in WT lacrimal glands in an IFNAR1-dependent but TLR7-independent manner enriched for clusters of pathways and processes of innate immune functions (Figure 5B, bottom). Together, these data indicate that TLR7 largely drives a type I IFN response but that type I IFN drives a separate set of innate immune genes in a TLR7-independent manner, while TLR7 drives B cell-related processes within the lacrimal glands in a type I IFN-independent manner.

## 3. Discussion

TLR7 has been implicated in multiple autoimmune diseases including SS and T1D [10,11,12,13]. As an X-linked gene, *TLR7* may escape X-linked inactivation leading to increased TLR7 levels that have been associated with increased systemic autoimmunity (such as SS) in individuals with more than one X chromosome [24,27,28]. Here, we evaluated the role of TLR7 in the spontaneous autoimmune manifestations in NOD mice by creating *Tlr7* KO NOD mice. We found that male NOD mice were protected from development of lacrimal gland inflammation and T1D development, while female NOD mice were not protected from developing salivary gland inflammation or T1D. This was not due to an X-linked phenomenon as TLR7 protein was not detected in either male or female *Tlr7* KO mice. Surprisingly, TLR7-deficiency resulted in development of salivary gland inflammation in some males, but no females (WT or *Tlr7* KO) developed lacrimal gland inflammation. These data suggest different disease mechanisms in the development of SS and T1D manifestations in male and female NOD mice with a role for TLR7 in promoting or protecting from disease manifestations in males but not females.

For the SS manifestations, we have previously demonstrated that male-specific lacrimal gland inflammation required intact type I IFN signaling [23]. In contrast, female salivary gland inflammation required intact type II IFN signaling as *Ifng* or *Ifngr* KO NOD mice failed to develop salivary gland inflammation [29]. Interestingly, male type II IFN signaling-deficient NOD mice were not protected from development of lacrimal gland inflammation [29], and female type I IFN signaling-deficient NOD mice were not protected from developing salivary gland inflammation (our unpublished observation). Thus, the two sex-dependent SS manifestations in NOD mice are dependent on different IFN signaling pathways. Studies in humans have demonstrated that a positive IFN-signature in individuals with SS may be dominated by type I IFN, type II IFN, or a combination of type I and type II IFN [30]. Together, these findings suggest that spontaneous SS-like disease in NOD mice may represent two different mechanisms of disease found in humans, with lacrimal gland disease in male NOD mice representing the type I IFN-dependent disease in humans and salivary gland disease in female NOD mice representing the type II IFN-dependent disease in humans. This is supported in part by recent demonstration of similarities in gene expression profiles in lacrimal glands of male NOD mice and salivary glands of humans, though that study did not specifically limit the human data to type I IFN-dominant disease [31].

TLR7 signaling is a well-established type I IFN-driving stimulus, which is in accordance with our findings of the requirement of TLR7 for male-specific lacrimal gland inflammation in NOD mice but not for female-specific salivary gland disease. In support of this, a large majority (>80%) of genes up-regulated in lacrimal glands of WT NOD mice in a TLR7-dependent manner overlap with those genes up-regulated in a type I IFN-signaling-dependent manner. These genes were enriched for pathways involved in the adaptive immune response. Interestingly, though, the smaller portion of genes up-regulated in a TLR7- but not IFNAR1-dependent manner enriched for pathways related to B cell responses suggesting a unique role for TLR7 signaling in B cell responses within lacrimal glands. In SS, B cells may play pathogenic roles through several mechanisms including production of autoantibodies, presentation of autoantigens to T cells, and production of inflammatory cytokines [32]. In B cell-deficient NOD mice, the initiation of lacrimal and salivary gland inflammation was not prevented [33]. However, in WT NOD mice, B cells are present within gland infiltrates and accumulate in the lacrimal glands over time [31,34]. Together with our data here, these studies suggest that B cells are not required for initiation of exocrine gland inflammation but may contribute to the continued inflammation within the glands in a TLR7-dependent manner. In support of a pathogenic role for TLR7 in lacrimal gland inflammation, male BXSB/MpJ-*Yaa* mice develop spontaneous lacrimal gland inflammation [9]. These mice express higher levels of TLR7 due to translocation of X chromosome genes including *Tlr7* to the Y chromosome resulting in increased TLR7 protein expression [35]. Notably, the lacrimal gland infiltrates in these mice were dominated by B cells [9]. In B cells from humans with SS, *TLR7* up-regulation has also been reported [16,36]. TLR7 is expressed by many cells beyond B cells, and additional studies are necessary to determine if the pathogenic role of TLR7 in driving lacrimal gland inflammation in NOD mice is B cell-intrinsic and/or dependent on other TLR7-expressing cells.

While the different IFN-dependencies in the sex-specific SS manifestations in NOD mice may explain the male-specific protection from lacrimal gland disease in *Tlr7* KO mice, this does not explain the sex differences in T1D protection. Studies of IFN signaling-deficient NOD mice suggested a more complex role for type I and II IFN signaling in T1D development. Female NOD mice lacking either *Ifnar1* or *Ifngr1* showed some (but incomplete) protection, but males were protected only when both IFN receptor genes were disrupted [22]. Notably, other studies reported no change in T1D development in female NOD mice deficient in type I IFN signaling [37] or type II IFN signaling [38], but these studies did not assess disease in males. In contrast to SS manifestations in NOD mice that occur only in one sex (lacrimal disease in males, salivary gland disease in females), T1D may develop in both males and females, though with different incidence generally favoring increased disease in females. The female T1D bias related to differences in microbiota and was lost when NOD mice were kept in germ free conditions [39,40]. MyD88-dependent signaling contributed to the role of microbiota on T1D development in NOD mice. Both male and female MyD88-deficient NOD mice were protected from diabetes but only when gut microbiota were present [41]. TLR7 signaling is MyD88-dependent, but our findings here demonstrate a role for TLR7 signaling in T1D development only in males suggesting other MyD88-dependent TLRs may compensate for the loss of TLR7 in females in our studies. TLR2-deficient female NOD mice were partially protected from T1D development, and this also required the presence of microbiota [42]. Both male and female TLR9-deficient NOD mice were protected from developing T1D [43]. Together these data suggest that multiple TLRs play pathogenic roles in T1D development in NOD mice, but these roles differ between sexes with TLR7 providing a non-redundant role in T1D development specifically in male NOD mice. Whether these sex differences in the role of TLR7 in T1D development also depend on microbiota remains to be determined.

To better understand the role of TLR7 in lacrimal gland autoimmunity, we performed RNA sequencing studies on lacrimal glands and identified both common and unique immune pathways up-regulated in TLR7 or type I IFN-signaling-dependent manners. TLR7-dependent (but type I IFN-independent) pathways suggested a role for TLR7 in driving B cell responses in the lacrimal glands. Given the role of TLR7 in B cell biology, it is not surprising that three of the top five DE genes in WT lacrimal glands were immunoglobulin-related genes. The remaining genes in the top 5 DE in WT lacrimal glands in a TLR7-dependent manner (*C4a* and *Il21*) also play roles in B cell responses. *C4a* encodes complement protein 4a, which plays a key role in both the classical and lectin pathways of the complement cascade [44]. Deficiency in complement C4 protein has been associated with systemic autoimmunity in humans including the recent finding that copy number variations in the *C4A* and *C4B* genes were associated with increased risk of SS [45]. In humans, though, a decrease in copy number resulting in lower protein levels was associated with higher risk of autoimmunity, which is contrary to our findings of an increase in *C4a* expression associated with lacrimal gland inflammation in WT mice. Since complement proteins are involved in the clearance of cell debris, the up-regulation of *C4a* may be a consequence of the epithelial cell damage in the context of inflammation rather than a cause in NOD mice, and the lack of C4 proteins due to decreased copy number variations in the associated genes in humans may contribute to ongoing inflammation by failing to adequately clear the debris. Given that NOD mice are deficient in C5, which is downstream of C4, they may have a similar deficiency in clearing cell debris through complement-mediated mechanisms [46]. Despite the association of deficiencies in C4 with autoimmunity, laser capture microdissection coupled with gene expression studies demonstrated an increase in *C4A* expression in ductal cells isolated from minor salivary glands of individuals with SS compared to healthy controls [47]. Together, these results suggest that C4 plays a role in the immune regulation and/or dysregulation in the development of SS autoimmunity, but additional studies are needed to further define these mechanisms.

Interleukin-21 (IL-21) is an inflammatory cytokine produced by innate and adaptive immune cells that promotes B and T cell responses and may affect innate immune and non-immune cell functions. The role of IL-21 in SS has been reviewed [48] and is further discussed in our recent report demonstrating decreased lacrimal gland inflammation in *Il21* KO NOD mice [26]. IL-21 may be produced by follicular helper CD4 T cells (Tfh), which are characterized by expression of chemokine receptor CXCR5 and transcription factor BCL6. In NOD lacrimal glands, another IL-21-producing CD4 population that lacks these classic Tfh markers has been identified [31]. This population of CD4 T cells expressed high levels of PD1, ICOS, CD73, and CD200 by flow cytometry, and transcriptional analyses demonstrated increased expression of *Tbx1*, *Tnf*, and *Ifng* suggesting a Th1-type profile. Among the genes up-regulated in these IL-21-producing CD4 T cells was *Sostdc1*, which we have also identified here as being highly DE in lacrimal glands of WT NOD mice in a TLR7-dependent manner. *Sostdc1* encodes Sclerostin domain-containing protein 1, a secreted protein that antagonizes both Wnt/β-catenin and bone morphogenetic protein signaling. *Sostdc1* plays many roles in development of tissues (including development of the eye [49]) and in promoting cancer. Recent study using SOSTDC1-reporter mice identified a potential regulatory role for Tfh-secreted SOSTDC1 in promoting the development of regulatory follicular T cells in germinal centers following viral infection or immunization [50]. Notably, these SOSTDC1-secreting Tfh cells expressed lower levels of *Il21* and lost the ability to help B cells. Whether the up-regulated *Sostdc1* in WT NOD lacrimal glands plays a pathogenic or regulatory role remains to be determined.

In summary, TLR7 is required for the spontaneous autoimmune manifestations that develop in WT male NOD mice including lacrimal gland inflammation and T1D development but is dispensable for the salivary gland inflammation and T1D that spontaneously develops in females. Moreover, TLR7 may provide a protective role in salivary gland disease in male NOD mice. In lacrimal glands, TLR7 is associated with many adaptive immune response genes common to a type I IFN response but also promotes B cell activity within the lacrimal glands in a type I IFN-independent manner. Previous microarray studies have identified increased *Tlr7* expression (~3-fold) in lacrimal glands from WT male compared to female NOD mice [51], which may be dependent on type I IFN signaling as *Tlr7* was expressed ~4-fold greater in WT male NOD lacrimal glands compared to *Ifnar1*-deficient male NOD lacrimal glands [26]. Thus, the pathogenic roles of TLR7 and type I IFN signaling in lacrimal gland autoimmunity in NOD mice are complex (Figure 6). Additional functional studies are required to determine the direct roles of TLR7 signaling in driving the gene expression changes reported here or, instead, to determine if additional complex interactions downstream of TLR7 are required. One key unanswered question that remains is what TLR7 ligands trigger the TLR7 response in lacrimal glands in the context of SS-like autoimmunity. While the possibility of an exogenous viral trigger for TLR7 signaling was not formally assessed, we favor the possibilities of either an endogenous retrovirus or an endogenous nonviral ligand released from dying epithelial or immune cells during lacrimal gland inflammation. In these regards, the recent identification of a new endogenous TLR7 ligand, U11snRNA [52], may warrant further study in NOD mice. Ultimately, identifying the innate signals that drive lacrimal gland inflammation will provide targets for novel diagnostic tests and therapeutic modalities.

## 4. Materials and Methods

### 4.1. Mice

Male and female NOD/ShiLtJ (NOD) mice were purchased from The Jackson Laboratory (Bar Harbor, ME) and bred in our colonies. The recently described NOD. *Ifnar1^em16/em16^* (*Ifnar1* KO) mice were bred in our colony [23]. Generation of NOD. *Tlr7*^−/−^ (*Tlr7* KO) mice were accomplished through targeting the *Tlr7* gene by CRSPR/Cas9-mediated gene editing. NOD mouse embryos were microinjected with 3 pL of a solution containing Cas9 mRNA and single guide RNA (sgRNA) at respective concentrations of 100 ng/mL and 50 ng/mL. The sgRNA sequence was designed to target ATTTACAGGTGTTTTCGATG in Tlr7. Genomic tail DNA was screened by Sanger sequencing. The genomic region around the targeted site was amplified by PCR with primers Tlr7-geneF (5′-ACTGACATATGCAAAGCATA-3′) and Tlr7-geneR (5′-ATTTCTTCCAGATGGTTCAGCCTA-3′). The resulting PCR product was purified and sequenced. During the screening of the N1 progeny, an allele with a 2 base-pair deletion was identified. The mouse was backcrossed to NOD for 1 generation, and the mutation was subsequently fixed to homozygosity. Male and female mice were used at ages ranging from 8–30 weeks. Mice were monitored for diabetes using Diastix urine glucose strips (Bayer Diagnostics, Whippany, NJ, USA). For diabetes studies, mice were monitored twice weekly and onset of diabetes was defined by two consecutive readings of >250 mg/dL. Mice were maintained in accordance with Institutional Animal Care and Use Committee Guidelines, and reported studies were approved by the Institutional Animal Care and Use Committees at the University of Iowa (0921655, approved 19 March 2019) and Medical College of Wisconsin (AUA1863, approved 23 April 2019).

### 4.2. Flow Cytometry

Spleens were removed and splenocytes dissociated using the end of a 3 mL syringe plunger through a 70 µm nylon mesh in Roswell Park Memorial Institute 1640 (RPMI) (Life Technologies, Waltham, MA, USA) supplemented with 10% fetal bovine serum, 100 U/mL penicillin, 100 µg/mL streptomycin and 50 µM β-mercaptoethanol (complete RPMI). Red blood cells were depleted through treatment with Ammonium-Chloride-Potassium (ACK) lysis buffer (Lonza, Mapleton, IL, USA). Single cell splenocyte suspensions were stained with fluorophore-conjugated monoclonal antibodies: CD11c (clone N418, eFluor450), B220 (clone RA3-6B2, PerCP-Cy5.5), TLR7 (clone A94B10, phycoerythrin (PE)), mouse IgG_1_, kappa isotype (PE), purchased from eBioscience (San Diego, CA, USA), BioLegend (San Diego, CA, USA), or BD Biosciences (San Jose, CA, USA). For intracellular staining for TLR7 (or isotype control), cells were fixed and permeabilized with the Foxp3/Transcription Factor Staining Buffer Set per manufacturer’s protocol (eBioscience). Flow cytometry data acquisition was performed on a BD LSR II (BD Biosciences) then analyzed with FlowJo software (Treestar Inc., Ashland, OR, USA). For analyses, cells were gated based on size and complexity (FSC-A by SSC-A) then to identify singlets (FSC-A by FSC-W). Singlets were subsequently gated on the CD11c^+^B220^+^ population to quantitate TLR7 expression in this population of cells.

### 4.3. Histology and Quantitation of Exocrine Gland Inflammation

Quantitation of lacrimal and salivary gland inflammation was performed as previously described [53]. Briefly, exorbital lacrimal glands and submandibular salivary glands were fixed in formalin, processed, embedded in paraffin, and 5 µm sections were stained with hematoxylin and eosin (H&E). Inflammation was quantified in a blinded manner by standard light microscopy at 10× objective using standard focus-scoring with a focus defined as an aggregate of at least 50 mononuclear cells and the focus score defined as the number of foci per 4 mm^2^ of tissue. Tissue areas were calculated by ImageJ software [54] using low magnification digital images obtained by scanning H&E-stained sections with the PathScan Enabler IV (Meyer Instruments, Houston, TX, USA). The WT lacrimal and salivary glands in Figure 1 were previously published in comparison to another KO NOD strain [21] from the same colony as the *Tlr7* KO samples to which they were compared in this study. Representative H&E-stained sections in the figure were obtained by whole-slide scans using the PathScan Enabler 5 (Meyer Instruments).

### 4.4. RNA Sequencing of Lacrimal Gland RNA and Bioinformatics Analyses

RNA was isolated from lacrimal glands per manufacturer’s protocol using RNeasy Plus Mini Kit (Qiagen, Valencia, CA, USA) and submitted to the Iowa Institute of Human Genetics (IIHG) Genomics Core Facility. Barcoded samples were pooled and sequenced using an Illumina HiSeq 4000. Reads were demultiplexed and converted from the native Illumina BCL format to fastq format using an in-house python wrapper to Illumina’s “bcl2fastq” conversion utility. FASTQ data were processed with “bcbio”, a best-practices pipeline available at the open-source “bcbio-nextgen” project (https://github.com/chapmanb/bcbio-nextgen; version 1.0.8) [55]. Reads were aligned to mm10 (genome FASTA and annotations derived from ftp://ftp.ensembl.org/pub/release-97/fasta/mus_musculus/) reference genomes using the ultra-rapid “hisat2” aligner (ver 2.1.0) [56]. Concurrently, reads were also quantified against the transcriptome using the “salmon” aligner (ver 0.9.1) [55], yielding estimated counts and values in length-normalized TPM (transcripts per million). Transcript-level abundances were converted to gene-level counts using the “tximport” package from Bioconductor [56]. Read and alignment quality control was performed with qualimap and samtools operating on the BAM alignments [57,58,59]. All samples passed quality control with ~70% of reads mapping, and ~70% of mapped reads mapping to exonic regions. Sequencing depth ranged from 48–70 M reads/sample. Inspection of the PCA plot and lacrimal gland focus scores led to dropping one outlier sample from the *Ifnar1* KO sample group (resulting in n = 3), which clustered with WT and had WT-level inflammation. No samples were dropped from the *Tlr7* KO (n = 5) or WT control (n = 3) groups. Gene-level counts were used for differential gene expression analysis with DESeq2 [60]. Prior to calculating DE genes, the counts table was filtered to exclude genes where at least two WT samples had a sum of counts less than 30. This was done to prevent extreme outlier zero counts (i.e., dropouts) within the WT replicates from creating very large fold-change artifacts. Data have been deposited in NCBI’s Gene Expression Omnibus [61] and are accessible through GEO Series accession number GSE161184 (https://www.ncbi.nlm.nih.gov/geo/query/acc.cgi?acc=GSE161184). DE gene expression data was analyzed using iPathwayGuide (Advaita Bioinformatics, https://www.advaitabio.com/ipathwayguide) to detect and predict significantly impacted pathways, biological processes, and molecular interactions. These analyses implement an “impact analysis” approach, which considers the direction and type of all signals on a pathway along with the position, role and type of each gene [25,62,63,64]. For analyses of enriched pathways and processes common to and unique to the two different RNA sequencing data sets (TLR7-dependent and IFNAR1-dependent), genes up-regulated at least 2-fold in WT compared to KO were analyzed with Metascape (http://metascape.org/gp/index.html#/main/step1) [65].

### 4.5. Quantitative PCR

Quantitative PCR (qPCR) was performed as previously described [23]. Briefly, lacrimal glands were isolated and stored in RNAlater (Invitrogen, Carlsbad, CA, USA). Samples were homogenized and total RNA isolated per manufacturer’s protocol using the RNeasy Plus Mini Kit (Qiagen). cDNA was generated with the SuperScript II Reverse Transcriptase Kit (Invitrogen) and random primers (Invitrogen). qPCR was performed using Power SYBR Green PCR Master Mix (Applied Biosystems, Foster City, CA, USA) and appropriate gene-specific primers (Table 5) on a QS-7 FLEX Real Time PCR System (Applied Biosystems). Number of gene transcripts were quantified based on standard curve generated with plasmid cDNA and normalized to housekeeping gene *Hk2*.

### 4.6. Statistical Analyses

Statistical analyses were performed with Prism 8.4.3 (GraphPad, San Diego, CA, USA) except for bioinformatics analyses using iPathwayGuide or Metascape as described above. Two-group comparisons of data approximating normal distribution (flow cytometry data) were performed by unpaired *t*-test with Welch’s correction for unequal variance where necessary. Two-group comparisons of non-normally distributed data (focus scores, qPCR) were performed by Mann-Whitney test. Diabetes incidence comparisons were performed by Log-rank test. *p* < 0.05 was considered significant.

## Figures and Tables

**Figure 1 ijms-21-09478-f001:**
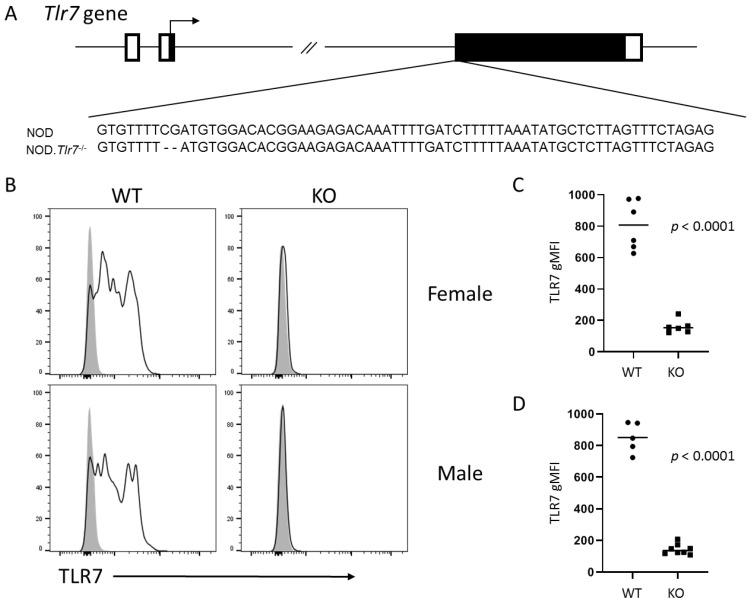
Generation and validation of *Tlr7* KO NOD mice. (**A**) Schematic of the *Tlr7* gene including the relevant sequence in wild-type NOD mice (NOD) and the 2 base-pair deletion mediated by CRISPR/Cas9 targeting of *Tlr7* in NOD embryos resulting in the *Tlr7* KO (NOD. *Tlr7*^−/−^) strain used in our studies. Boxes represent exons with the filled portions representing the coding regions. (**B**) Representative flow cytometry histograms demonstrating lack of TLR7 protein in splenocytes of wild-type (WT) and *Tlr7* KO (KO) NOD females (top) and males (bottom) as indicated. Splenocytes were permeabilized for intracellular staining with a TLR7-specific (solid black lines) or an isotype control (filled gray histograms) antibody (*x*-axis). Plots are gated on CD11c^+^B220^+^ singlets and are normalized to mode (*y*-axis). (**C**,**D**) Cumulative quantitation (geometric mean fluorescence intensity, gMFI) of TLR7 expression in cells represented in (**B**) from female (**C**) or male (**D**) WT or *Tlr7* KO NOD mice. Symbols represent individual mice (5–8 per group), lines are means. *p*-value by unpaired *t*-test with Welch’s correction for unequal variance where necessary.

**Figure 2 ijms-21-09478-f002:**
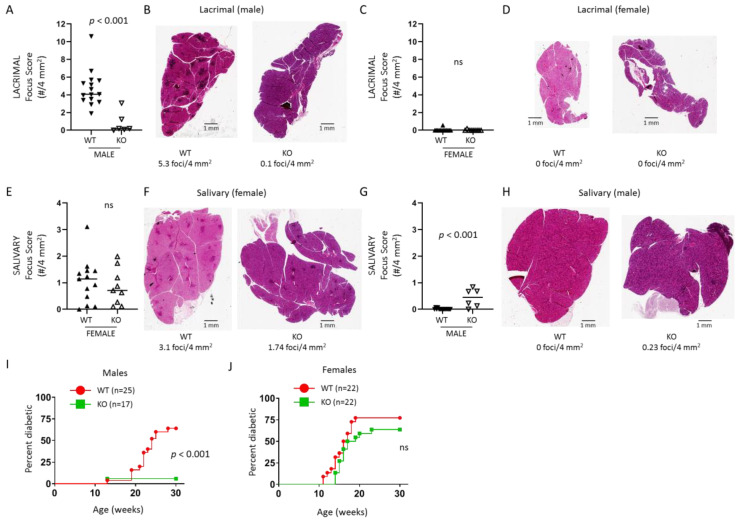
Male *Tlr7* KO NOD mice are protected from lacrimal gland autoimmunity and T1D development. (**A**,**C**) Quantitation of lacrimal gland inflammation in 10-week-old WT or *Tlr7* KO male (**A**) or female (**C**) mice. Focus score equals number of inflammatory foci (minimum of 50 mononuclear cells per focus) per 4 mm^2^. Symbols represent individual mice (males: n = 15 WT, n = 6 *Tlr7* KO; females: n = 13 WT, n = 9 *Tlr7* KO), lines are medians. *p*-value by Mann-Whitney test. (**B**,**D**) Representative hematoxylin and eosin (H&E) stained sections of lacrimal glands from (**A** or **C**) as indicated below each image. Scale bars are 1 mm. (**E**,**G**) Quantitation of sialadenitis in 10-week-old WT or *Tlr7* KO female (**E**) or male (**G**) mice. Symbols, lines, scale bars, and *p*-value as in (**A**). ns, not significant. (**F**,**H**) Representative H&E-stained sections of salivary glands from (**E** or **G**) as indicated. (**I**,**J**) T1D incidence of male (**I**) and female (**J**) WT and *Tlr7* KO mice with indicated *p*-value by Log-rank test. ns, not significant.

**Figure 3 ijms-21-09478-f003:**
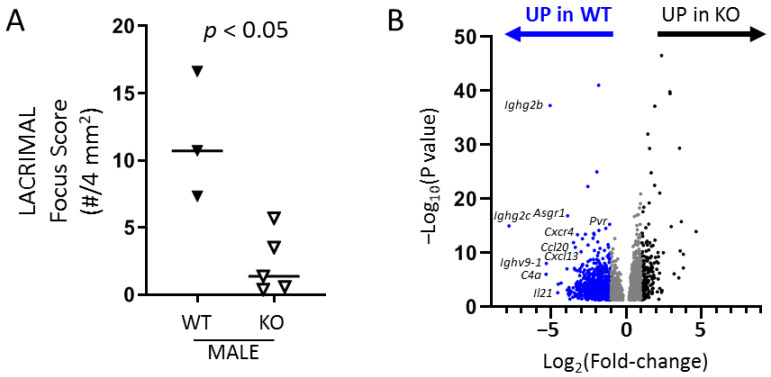
Identification of DE genes up-regulated in lacrimal glands of WT compared to *Tlr7* KO NOD mice. Lacrimal glands were isolated from 20–22-week-old WT (n = 3) or *Tlr7* KO (n = 5) NOD mice. One gland from each mouse was prepared for histological analyses to quantify inflammation. The other gland was used for RNA sequencing studies. (**A**) Quantitation of lacrimal gland inflammation by focus-scoring as in Figure 1. Symbols represent individual mice, lines are medians. *p*-value by Mann-Whitney test. (**B**) Volcano plot of DE genes in lacrimal glands of NOD mice with genes up-regulated in WT or KO as indicated above the graph. Dots represent individual genes with blue indicating at least 2-fold up-regulation in WT and black indicating at least 2-fold up-regulation in KO. Gray genes are up-regulated with fold-change less than 2. Select genes up-regulated in WT are indicated to the left of their respective data points (except *Ighg2c*, which is above the data point).

**Figure 4 ijms-21-09478-f004:**
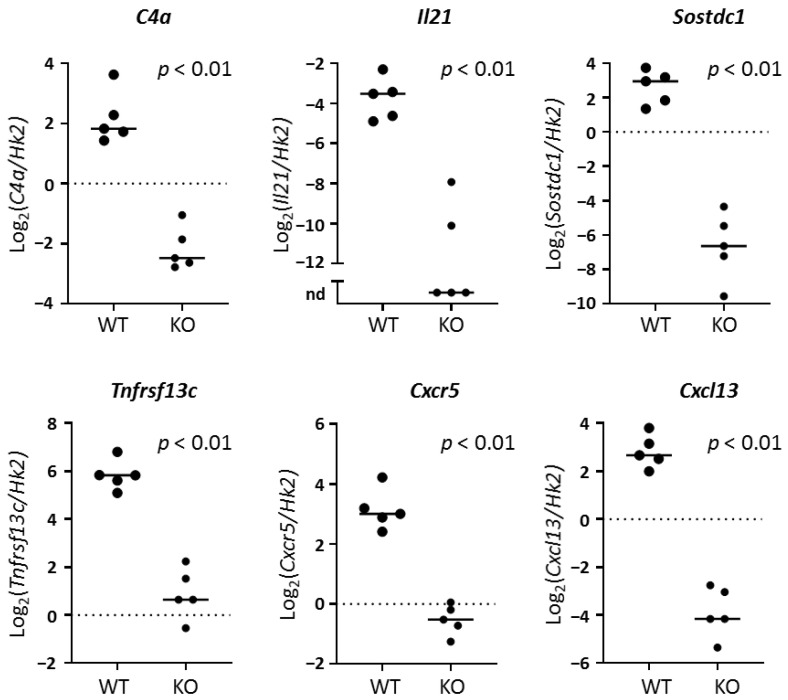
Validation of immune genes up-regulated in WT compared to *Tlr7* KO lacrimal glands. Graphs depict log-transformed relative expression of indicated genes normalized to housekeeping gene *Hexokinase 2* (*Hk2*) in lacrimal glands of 20–21-week-old male WT (n = 5) or *Tlr7* KO (n = 5) NOD mice (*x*-axis). Symbols represent individual mice. Solid lines are medians. Broken lines represent *y*-axis zero-value for reference given the different *y*-axis ranges. *p*-values by Mann-Whitney test. nd, not detected.

**Figure 5 ijms-21-09478-f005:**
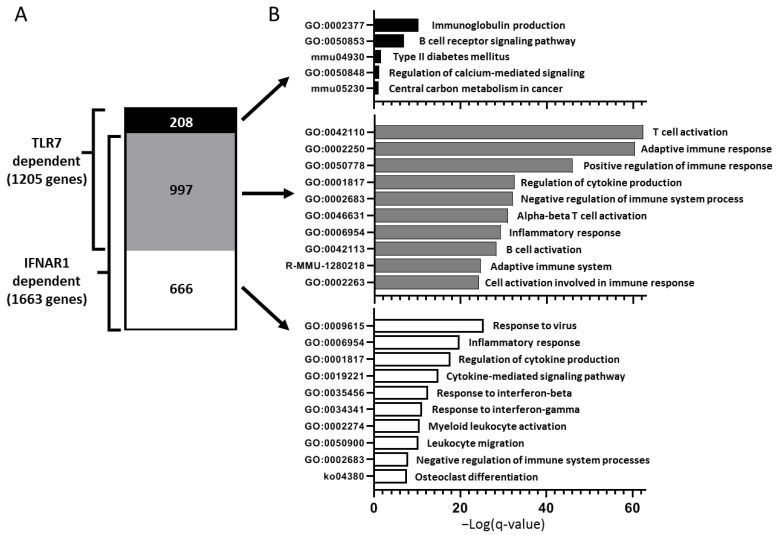
Overlapping and unique roles for TLR7 and IFNAR1 signaling in lacrimal gland inflammation. (**A**) Vertical slices graph of DE genes up-regulated at least 2-fold in lacrimal glands of WT compared to either *Tlr7* KO (TLR7-dependent) or *Ifnar1* KO (IFNAR1-dependent) NOD mice demonstrating those genes unique to the TLR7 data set (black), common to both data sets (gray), or unique to the IFNAR1 data set (white). (**B**) Bar graphs depict the enrichment clusters of processes and pathways (GO, KEGG, Reactome) based on genes from the three different categories in (**A**) as indicated. *X*-axes represent the negative log of the q-value, calculated using Benjamini–Hochberg procedure, and are on the same scale as indicated at the bottom.

**Figure 6 ijms-21-09478-f006:**
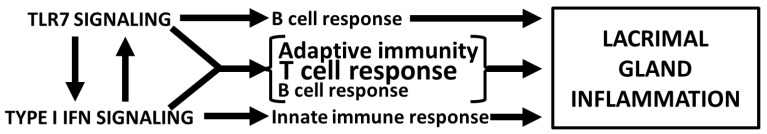
Complex roles of TLR7 and type I IFN in the development of lacrimal gland inflammation. TLR7 signaling drives a type I IFN response, and type I IFN signaling is associated with increased *Tlr7* expression in male NOD mouse lacrimal glands. Each of these signals also drives different aspects of the immune response including TLR7 driving a B cell response and type I IFN driving an innate immune response. TLR7 and type I IFN signals together drive a more robust adaptive immune response dominated by a T cell response. Together these signals lead to focal lymphocytic inflammation within lacrimal glands that characterizes SS.

**Table 1 ijms-21-09478-t001:** Top 20 DE genes up-regulated in lacrimal glands of WT mice compared to *Tlr7* KO mice ^1^.

Protein Name	Gene Symbol	Ensembl ID	LogFC	*p*-Value ^2^
Immunoglobulin heavy constant gamma 2C	*Ighg2c*	ENSMUSG00000076612	−7.82263	9.82 × 10^−16^
Complement 4a	*C4a*	ENSMUSG00000015451	−5.34572	8.51 × 10^−7^
Immunoglobulin heavy variable 9-1	*Ighv9-1*	ENSMUSG00000096805	−5.33515	8.83 × 10^−9^
Immunoglobulin heavy constant gamma 2B	*Ighg2b*	ENSMUSG00000076613	−5.07854	4.75 × 10^−38^
Interleukin-21	*Il21*	ENSMUSG00000027718	−4.57119	0.002267
Glutamate decarboxylase 1	*Gad1*	ENSMUSG00000070880	−4.54398	5.80 × 10^−5^
RIKEN cDNA G530011O06 gene	*G530011O06Rik*	ENSMUSG00000072844	−3.95434	8.40 × 10^−8^
histocompatibility 2, O region beta locus	*H2-Ob*	ENSMUSG00000041538	−3.94349	0.000705
Asialoglycoprotein receptor 1	*Asgr1*	ENSMUSG00000020884	−3.90959	1.26 × 10^−17^
Junctional cadherin complex regulator	*Jhy*	ENSMUSG00000032023	−3.86272	0.003172
Receptor-type tyrosine-protein phosphatase V	*Ptprv*	ENSMUSG00000097993	−3.81638	0.006741
Sclerostin domain containing 1	*Sostdc1*	ENSMUSG00000036169	−3.81371	0.000239
Mitogen-activated protein kinase kinase kinase 19	*Map3k19*	ENSMUSG00000051590	−3.78867	0.001609
B cell activating factor receptor (BAFF-R)	*Tnfrsf13c*	ENSMUSG00000068105	−3.77041	0.00071
Alpha-1-antitrypsin 1-1	*Serpina1a*	ENSMUSG00000066366	−3.76237	0.032582
Hepatitis A virus cellular receptor 1	*Havcr1*	ENSMUSG00000040405	−3.72396	0.003191
C-X-C motif chemokine receptor 5	*Cxcr5*	ENSMUSG00000047880	−3.63713	0.001409
carbohydrate sulfotransferase 3	*Chst3*	ENSMUSG00000057337	−3.53362	0.000264
Fc receptor-like 1	*Fcrl1*	ENSMUSG00000059994	−3.52988	0.000323
Polyunsaturated fatty acid (12S)/(13S)-lipoxygenase, epidermal-type	*Alox12e*	ENSMUSG00000018907	−3.51249	1.13 × 10^−12^

^1^ DE, differentially expressed; FC, fold-change (KO relative to WT); KO, knockout; WT, wild-type. ^2^
*p*-value adjusted for multiple comparisons by Benjamini–Hochberg procedure.

**Table 2 ijms-21-09478-t002:** Pathways enriched in WT lacrimal glands **^1^**.

Pathway	DE Genes/Total	*p*-Value ^2^
Cell adhesion molecules	66/101	4.34 × 10^−10^
Cytokine-cytokine receptor interaction	76/139	3.63 × 10^−6^
Antigen processing and presentation	39/63	3.63 × 10^−6^
Phagosome	71/131	1.04 × 10^−5^
Human T cell leukemia virus 1 infection	93/200	7.06 × 10^−5^
Natural killer cell mediated cytotoxicity	41/77	0.000132
Complement and coagulation cascades	27/45	0.000173
Leukocyte transendothelial migration	46/88	0.000248
Hematopoietic cell lineage	41/69	0.000349
Viral myocarditis	40/56	0.000494

^1^ DE, differentially expressed, WT, wild-type. ^2^
*p*-value with Bonferroni correction for multiple comparisons.

**Table 3 ijms-21-09478-t003:** Top DE genes in cell adhesion molecules pathway ^1^.

Gene	Ensembl ID	LogFC	*p*-Value ^2^
*H2-Ob*	ENSMUSG00000041538	−3.94349	0.000705275
*Cd22*	ENSMUSG00000030577	−3.31601	0.000119483
*Glycam1*	ENSMUSG00000022491	−3.13053	0.01223442
*Pdcd1*	ENSMUSG00000026285	−3.08293	2.57387 × 10^−6^
*H2-Oa*	ENSMUSG00000024334	−3.06688	0.000401944
*Sell*	ENSMUSG00000026581	−2.80719	0.008678722
*Cd2*	ENSMUSG00000027863	−2.76434	0.010163565
*H2-DMb2*	ENSMUSG00000037548	−2.76392	0.001751811
*H2-Q7*	ENSMUSG00000060550	−2.59737	0.018849387
*Ctla4*	ENSMUSG00000026011	−2.56514	0.010040012

^1^ DE, differentially expressed; FC, fold-change (KO relative to WT). ^2^
*p*-value with Bonferroni correction for multiple comparisons.

**Table 4 ijms-21-09478-t004:** Top DE genes in cytokine-cytokine receptor interaction pathway ^1^.

Gene	Ensembl ID	LogFC	*p*-Value ^2^
*Il21*	ENSMUSG00000027718	−4.57119	0.002267
*Tnfrsf13c*	ENSMUSG00000068105	−3.77041	0.00071
*Cxcr5*	ENSMUSG00000047880	−3.63713	0.001409
*Ccl20*	ENSMUSG00000026166	−3.39023	0.000001
*Cxcr4*	ENSMUSG00000045382	−3.24664	0.000001
*Cxcl13*	ENSMUSG00000023078	−3.00501	0.000001
*Ccr6*	ENSMUSG00000040899	−2.91986	7.71 × 10^−5^
*Tnfsf8*	ENSMUSG00000028362	−2.73458	0.00033
*Ccl19*	ENSMUSG00000071005	−2.70432	0.02331
*Ltb*	ENSMUSG00000024399	−2.64806	0.002418

^1^ DE, differentially expressed; FC, fold-change (KO relative to WT). ^2^
*p*-value with Bonferroni correction for multiple comparisons.

**Table 5 ijms-21-09478-t005:** Primer sequences for qPCR analyses.

Target	Sequence	References
*Tnfrsf13c* Fwd	AGATGGGCATGGTGGTACACA	[66]
*Tnfrsf13c* Rev	TGGAACTTGCTATGTAGACCAGGAT	
*Il21* Fwd	GGACAGTGGCCCATAAATCA	[62]
*Il21* Rev	CAGGGTTTGATGGCTTGAGT	
*Cxcr5* Fwd	TGGCCTTCTACAGTAACAGCA	[63]
*Cxcr5* Rev	GCATGAATACCGCCTTAAAGGAC	
*Sostdc1* Fwd	CACCCTGAATCAAGCCAGGA	[64]
*Sostdc1* Rev	TAGCCTCCTCCGATCCAGTT	
*C4a* Fwd	AAACTCAGAATCTTCAGCACAA	
*C4a* Rev	GGAAGAAAGATGGCTCTTTTG	
*Cxcl13* Fwd	CATAGATCGGATTCAAGTTACGCC	[67]
*Cxcl13* Rev	TCTTGGTCCAGATCACAACTTCA	
*Hk2* Fwd	CCCTGTGAAGATGTTGCCCAC	[68]
*Hk2* Rev	TGCCCATGTACTCAAGGAAGT	

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
