# Peer review of "Toll-Like Receptor 7 Is Required for Lacrimal Gland Autoimmunity and Type 1 Diabetes Development in Male Nonobese Diabetic Mice"

_ijms, 2020, doi:10.3390/ijms21249478_

Round 1

Reviewer 1 Report

The authors has made a interesting observation about the similarity of the autoimmune manifestation between SS and NOD. The approach and experimental design like the KO animal generation, transcriptomics were very appropriate for the manuscript. But the manuscript needs more functional experiments either in vitro or in vivo from the transcriptomics data to substantiate the findings, otherwise the manuscript will come across to the readers as little incomplete.

It will be good if the authors include the Ifnar1 KO mice data also in this manuscript, since some conclusions were derived from those animal model.

In the Materials and method, please write in more detail on the animal section.

It will be easy for the readers if the discussion is much more concise.

A schematic diagram explaining your findings in the conclusion section will be very useful for the readers to understand quickly.

Author Response

REVIEWER 1

The authors has made a interesting observation about the similarity of the autoimmune manifestation between SS and NOD. The approach and experimental design like the KO animal generation, transcriptomics were very appropriate for the manuscript. But the manuscript needs more functional experiments either in vitro or in vivo from the transcriptomics data to substantiate the findings, otherwise the manuscript will come across to the readers as little incomplete.

RESPONSE: We appreciate the reviewer’s comments and agree that additional functional studies (in vitro and in vivo) are necessary to further define the specific molecular mechanisms of the effects of TLR7 signaling in the development of these autoimmune manifestations. However, we have not yet performed these studies and, given the complexities of the system, expect that extensive studies will be needed to determine direct effects of TLR7 versus effects that may develop several steps downstream of TLR7 signaling. These effects may also be mediated within the same cell or with interactions between different cell types (including immune cells and epithelial cells). Because of these complexities, we will not be able perform satisfactory studies to address this in a timely manner. We added a statement (page 11, lines 354-356: Additional functional studies are required to determine the direct roles of TLR7 signaling in driving the gene expression changes reported here or, instead, to determine if additional complex interactions downstream of TLR7 are required.) to specifically note the importance of those functional studies to further understanding the role of TLR7. We feel that even without such studies, our main findings: 1) TLR7 is required for lacrimal gland autoimmunity, 2) identification of specific genes and pathways upregulated in lacrimal glands of WT compared to Tlr7 KO NOD mice, 3) similarities and differences in such upregulated genes and pathways when comparing to a similar WT vs Ifnar1 KO gene expression data set, are each valid and worthy of reporting regardless of the outcome of such functional studies.

It will be good if the authors include the Ifnar1 KO mice data also in this manuscript, since some conclusions were derived from those animal model.

RESPONSE: We have added information about the Ifnar1 KO mice in the methods. Supplemental Table 5 includes the list of the Ensembl gene IDs for the DE genes upregulated (at least 2-fold) in WT lacrimal glands compared to either Tlr7 KO or Ifnar1 KO lacrimal glands (as depicted in Figure 5). The addition of a full description and validation of the RNA sequencing data from the WT vs Ifnar1 KO lacrimal glands would add a substantial amount of data, additional figures and tables, and additional results and discussion text. Because of this, we are preparing a separate manuscript to describe the Ifnar1 KO mouse gene expression studies and feel it is better suited as a separate manuscript due to the already overwhelming amount of data to report based on this Tlr7 study alone. We hope to have that manuscript submitted very soon so that both will be available for side-by-side comparison for anyone who wishes to consider both RNAseq data sets together. Of note, though, the data sets were submitted together to GEO (GSE161184) and will be released and publicly available shortly after publication of these manuscripts.

In the Materials and method, please write in more detail on the animal section.

RESPONSE: We have added information regarding the Ifnar1 KO mice (referencing our prior study describing their generation) and specific details about regulatory approval of our studies to the Materials and Methods section 4.1. (page 11).

It will be easy for the readers if the discussion is much more concise.

RESPONSE: We appreciate this comment by the reviewer. This study (and all RNAseq or other big data studies) includes considerable data from which we hoped to describe some key findings. While our discussion is by no means exhaustive of such findings, we felt it important to discuss the main features in the context of the literature. Because our findings include effects related to disease manifestations (salivary glands, lacrimal glands, T1D) that occur independently, we felt the need to include discussion of each separately. And, we felt it was important to also include some discussion of the most DE genes relevant to Sjögren syndrome based on other studies. With this thinking, we cannot easily modify the discussion to make it more concise. If the reviewers or editors have specific suggestions for text to be removed from the discussion, we would be happy to consider this further.

A schematic diagram explaining your findings in the conclusion section will be very useful for the readers to understand quickly.

RESPONSE: We appreciate this suggestion and agree that schematic diagrams can be very helpful. We have added such a schematic to very broadly summarize the complexities of the roles of TLR7 and type I IFN signaling in the inflammatory process in NOD mouse lacrimal glands. (Figure 6 and text in discussion, page 11).

Reviewer 2 Report

In this paper by Debreceni et al, the authors evaluated the role of Tlr7 in the development of Sjøgren's disease like features in NOD mice by generating a Tlr7-/- mice model. The paper is well-written, easy to understand and the results are of interest.  

I have some few comments though.

-First, I kind of feel that you involve "apples and pears" here (both females and male mice) and then you proceed with only males, and you draw conclusions that this "drives the sex-differences". I like the design, but I had wished that you also RNA sequences the lacrimal gland in female mice, and also that you had sequenced the salivary gland in both male KO mice, female KO mice and then controls from both sexes. Otherwise, you are not analysing the complete picture, and you cannot directly draw the conclusion that this drives the male-specific manifestations but not the female ones. I do see your point, and you have a really nice discussion where you include work from others which, together with your results, might indicate your conclusion. I still believe in your data, but wish for a milder conclusion and perhaps a milder title.

-I do believe that you should write "Tlr7" and not "TLR7". This is not the human protein.

-Graphs in Fig. 1C and D: Please write "gMFI Tlr7" on the scale.

-Have you or others measured the Tlr7-expression in lacrimal glands from KO mice and different forms of NOD mice?

Author Response

REVIEWER 2

-First, I kind of feel that you involve "apples and pears" here (both females and male mice) and then you proceed with only males, and you draw conclusions that this "drives the sexdifferences".

RESPONSE: We appreciate these comments and how the original manuscript may have over-stated the role of TLR7 in driving sex differences based on the lack of a complete set of transcriptomic data from all organs from each sex. We apologize for this and have amended the text and title to remove that statement or any indication that TLR7 drives the sex differences. In the original manuscript, we had included analyses of the specific glands that are spontaneously affected in the development of Sjögren’s manifestations in NOD mice – that is, lacrimal glands in males and salivary glands in females. To be more inclusive, we have now added panels to Figure 2 showing the lacrimal gland inflammation scores for females (new Figure 2C, D) and the salivary gland inflammation scores for males (new Figure 2G, H). With these additions, we believe it is clear that WT female-specific disease is mirrored in the TLR7 KO mice (spontaneous salivary gland inflammation and T1D but no lacrimal gland inflammation in either WT or Tlr7 KO females), but the WT male-specific disease is not at all mirrored in the Tlr7 KO mice (KO develop significantly less lacrimal gland inflammation and less T1D but actually develop more salivary gland inflammation, which does not develop in WT males). While this does not simplify our findings, we feel it supports the notion that TLR7-deficiency alters the autoimmune manifestations in males but not females, based on measures of lacrimal and salivary gland inflammation and development of diabetes. We have added text to describe these new panels in Results Section 2.2 and Discussion (pages 8-9, lines 232-236).

I like the design, but I had wished that you also RNA sequences the lacrimal gland in female mice, and also that you had sequenced the salivary gland in both male KO mice, female KO mice and then controls from both sexes. Otherwise, you are not analysing the complete picture, and you cannot directly draw the conclusion that this drives the male-specific manifestations but not the female ones.

RESPONSE: Again, we appreciate the Reviewer’s comment here and have amended the text (as noted above) to indicate a more accurate interpretation of the data. While we agree that a bigger set of comparisons to include both organs from both sexes would have provided for some interesting comparisons that may have supported a role for TLR7 driving the male-specific manifestations, we unfortunately did not do such studies and, thus, cannot add such data at this time. However, given the addition of the Figure 2 panels described above, we feel that the data overall do strongly suggest a role for TLR7 modulating autoimmunity in males to a greater degree than in females. Regardless, we have removed statements that TLR7 drives the sex-specific differences.

I do see your point, and you have a really nice discussion where you include work from others which, together with your results, might indicate your conclusion. I still believe in your data, but wish for a milder conclusion and perhaps a milder title.

RESPONSE: We thank the reviewer for their kind and supportive comments and amended the conclusion and title to make them both milder.

-I do believe that you should write "Tlr7" and not "TLR7". This is not the human protein.

RESPONSE: We strive to be precise in our nomenclature and appreciate the reviewer’s comments regarding the differences in annotating human and mouse proteins. This actually came up in our writing of the original manuscript prompting a search for guidance on this exact matter. What we found was that genes are annotated differently between human and mouse with human being capitalized (eg, TLR7) but mouse genes including only first letter capitalization (eg, Tlr7). However, it seems that this difference is limited to genes and that the convention is for proteins to be written without italics and in all capital letters regardless of human or mouse (eg, both are TLR7). While we were not able to find a definitive source for this and we could not find specific information on the MDPI publisher’s website, we did find other journals with specific instructions that we chose to follow (eg, https://www.jci.org/kiosk/publish/genestyle). If the reviewers or editors have a better source to correct us on this, or if the publishers/editors have specific preferences, we will gladly comply. Otherwise, at this time based on what we have found regarding this matter, we respectfully disagree with the Reviewer’s suggestion that mouse and human protein designations should be noted differently.

-Graphs in Fig. 1C and D: Please write "gMFI Tlr7" on the scale.

RESPONSE: We added TLR7 to the gMFI y-axis label to make the graphs easier to interpret. Thanks for the suggestion.

-Have you or others measured the Tlr7-expression in lacrimal glands from KO mice and different forms of NOD mice?

RESPONSE: In other studies, we have found increased expression of Tlr7 in WT male compared to WT female lacrimal glands as well as in WT male compared to Ifnar1 KO male lacrimal glands. (Statements and references added on page 11, lines 349-352). We have not specifically looked for expression differences in lacrimal glands of WT versus Tlr7 KO NOD mice – on a gene expression level, it is not clear whether a difference or lack of difference would be relevant or expected since the mRNA may still be expressed at normal levels (but missing the 2 deleted bases in the KO) without protein expression. On a protein level, we have included protein expression measurements by flow cytometry (Figure 1). While this was measured using cells isolated from spleens, we have no reason to expect that protein expression that was not detected in Tlr7 KO spleen cells would be detectable in cells isolated from Tlr7 KO lacrimal glands as this is a global KO model. However, we plan for future such flow cytometric analyses using cells isolated from WT lacrimal glands to help gain a better understanding of the role of TLR7 in contributing to the inflammatory response in the context of lacrimal gland autoimmunity. It might be more relevant to do such analyses comparing WT to Ifnar1 KO mice, which should both have the capabilities of expressing TLR7, but we can include Tlr7 KO mice as well in those studies.

Round 2

Reviewer 1 Report

Thank you for the explanations and the corrections made.